# Observation of continuum Landau modes in non-Hermitian electric circuits

Xuewei Zhang[1,2,6], Chaohua Wu [1,6], Mou Yan[1,3], Ni Liu[4], Ziyu Wang [5] ✉ & Gang Chen [1,2] ✉

Continuum Landau modes − predicted recently in a non-Hermitian Dirac Hamiltonian under a uniform magnetic field − are continuous bound states with no counterparts in Hermitian systems. However, they have still not been confirmed in experiments. Here, we report an experimental observation of continuum Landau modes in non-Hermitian electric circuits, in which the non-Hermitian Dirac Hamiltonian is simulated by non-reciprocal hoppings and the pseudomagnetic field is introduced by inhomogeneous complex on-site potentials. Through measuring the admittance spectrum and the eigenstates, we successfully verify key features of continuum Landau modes. Particularly, we observe the exotic voltage response acting as a rainbow trap or wave funnel through full-field excitation. This response originates from the linear relationship between the modes' center position and complex eigenvalues. Our work builds a bridge between non-Hermiticity and magnetic fields, and thus opens an avenue to explore exotic non-Hermitian physics.

As the fundamental and important issue in modern physics, charged particles subject to a uniform magnetic field have motivated the discovery of many intriguing phenomena, such as integer and fractional quantum Hall effects[1–4]. These phenomena are rooted in the generation of discrete Landau levels in the energy spectrum of the system. The separated Landau level modes are highly degenerate and spatially localized. One of the most striking examples manifesting this effect is the Dirac particle with linear dispersion under a magnetic field, who are generally mimicked in the graphene near the Dirac cones[5–7]. Such separated Landau levels have been realized in electric system[8], as well as photonic[9], acoustic[10–12] and mechanical[13,14] metamaterials.

In parallel, non-Hermitian systems, typically including complex on-site potential and non-reciprocal hopping, have attracted considerable attention. On the one hand, they host many unique properties, such as complex energy spectra[15], exceptional points and rings[16–19], and skin effect[20–27], which have no counterparts in Hermitian systems. Moreover, these unconventional properties bring potential applications in sensing[28–30], lasing[31,32], and wave manipulation[33–36].

Recently, the interplay of non-Hermiticity and magnetic field has generated a phenomenon termed as continuum Landau modes (CLMs)[37]. The CLMs have Gaussian spatial envelopes and form a continuous spectrum filling the complex energy plane. Remarkably, these CLMs violate the intuition in Hermitian systems where the bound states should be quantized. However, such important CLMs have still not been confirmed in experiments.

In this paper, we report an experimental observation of the CLMs in non-Hermitian electric circuit networks. In our system, the non-Hermitian Dirac Hamiltonian is simulated by non-reciprocal hoppings and a pseudomagnetic field is introduced by inhomogeneous complex on-site potentials. By measuring the circuit admittance spectrum and eigenstates, the key features of the CLMs are successfully observed. Intriguingly, we visualize an exotic steady-state voltage response through full-field excitation. This response results from the linear relationship between the CLMs' center position and complex eigenvalues. We also observe the CLMs in two types of 1D non-Hermitian electric circuits, exhibiting the unique rainbow trap and wave funnel.

[1]School of Physics and Microelectronics, Key Laboratory of Materials Physics of Ministry of Education, Zhengzhou University, Zhengzhou 450001, China. [2]State Key Laboratory of Quantum Optics and Quantum Optics Devices, Institute of Laser spectroscopy, Shanxi University, Taiyuan 030006, China. [3]Institute of Quantum Materials and Physics, Henan Academy of Sciences, Zhengzhou 450046, China. [4]Institute of Theoretical Physics, Shanxi University, Taiyuan 030006, China. [5]The Institute of Technological Sciences, Wuhan University, Wuhan 430072, China. [6]These authors contributed equally: Xuewei Zhang, Chaohua Wu. ✉e-mail: zywang@whu.edu.cn; chengang971@163.com

Our work provides a fertile platform to study exotic non-Hermitian physics driven by magnetic fields.

## Results

### Continuum Landau modes in non-Hermitian circuit lattice

Figure 1a shows our considered 2D non-Hermitian square lattice with both the non-reciprocal hopping and inhomogeneous complex on-site potential. To be concrete, this model consists of reciprocal hoppings $t_x$ along the $x$ direction, together with non-reciprocal hoppings $\pm t_y$ along the $y$ direction. Moreover, the sites in the $x$ and $y$ directions are respectively subjected to linear imaginary and real potentials, $-imB_x$ (the color of the dots) and $nB_y$ (the size of the dots), where $m$ and $n$ denote the site indices, and $B_{x,y}$ is the strength of the linear potential. Figure 1b shows a part of the designed circuit structure for emulating such lattice. The reciprocal hopping is achieved via capacitors $C_1$, while the non-reciprocal hoppings are implemented through an impedance converter with current inversion (INIC) of capacitance $\pm C_2$[26]. Furthermore, the linear real and imaginary potentials are realized by grounding the nodes with position-dependent capacitors $nC_0$ and resistors $R_0/m$, respectively.

According to the Kirchhoff rule, the response of a circuit system is described by $I = JV$, where $J$ is the admittance matrix or circuit Laplacian, and the vector components of $I$ and $V$ correspond to the input currents and voltages at the nodes in the circuit, respectively. For a given a.c. frequency $\omega = 2\pi f$, the circuit Laplacian can be expressed as

(see Supplementary Note 1 for details)

$$J(\omega)/(i\omega) = \sum_{m,n} \left[ t_x(|m,n\rangle\langle m+1,n| + \text{H.c.}) + t_y(|m,n\rangle\langle m,n+1| - \text{H.c.}) \right.$$
$$\left. + \left( nB_y - imB_x + \epsilon_0 \right)|m,n\rangle\langle m,n| \right],$$
(1)

where $t_x = -C_1$, $t_y = -C_2$, $B_y = C_0$, $B_x = 1/(\omega R_0)$, $\epsilon_0 = C_1 - 1/(\omega^2 L_0)$, and H.c. represents the Hermitian conjugate. Note that only the parameters $B_x$ and $\epsilon_0$ can be controlled by the frequency.

In the continuum limit and slowly varying envelope approximation (see Supplementary Note 2 for details), Eq. (1) becomes $J_\mathbf{k}/(i\omega) = \mathcal{E}_\mathbf{k}^0 - (-i\mu_\mathbf{k}\partial_x - B_y y) + i(-i\nu_\mathbf{k}\partial_y - B_x x)$, where $\mathcal{E}_\mathbf{k}^0 = \epsilon_0 + 2t_x \cos k_x - i2t_y \sin k_y$, $\mu_\mathbf{k} = 2t_x \sin k_x$, $\nu_\mathbf{k} = -2t_y \cos k_y$, and $k_{x,y} \in [-\pi, \pi]$. Remarkably, $J_\mathbf{k}/(i\omega)$ is an analog of the non-Hermitian Dirac Hamiltonian with complex linear dispersion[38,39] under the symmetric-gauge pseudovector potential $\mathbf{A} = (-B_y y, B_x x)$, which corresponds to a uniform pseudomagnetic field $\mathbf{B} = \nabla \times \mathbf{A} = (B_x + B_y)\hat{z}$. Clearly, the non-Hermitian Dirac Hamiltonian is simulated by non-reciprocal hoppings, while the pseudomagnetic field is introduced by inhomogeneous complex on-site potentials and can be controlled by frequency.

The introduction of the pseudomagnetic field in such no-Hermitian Dirac Hamiltonian may generate phenomenon such as

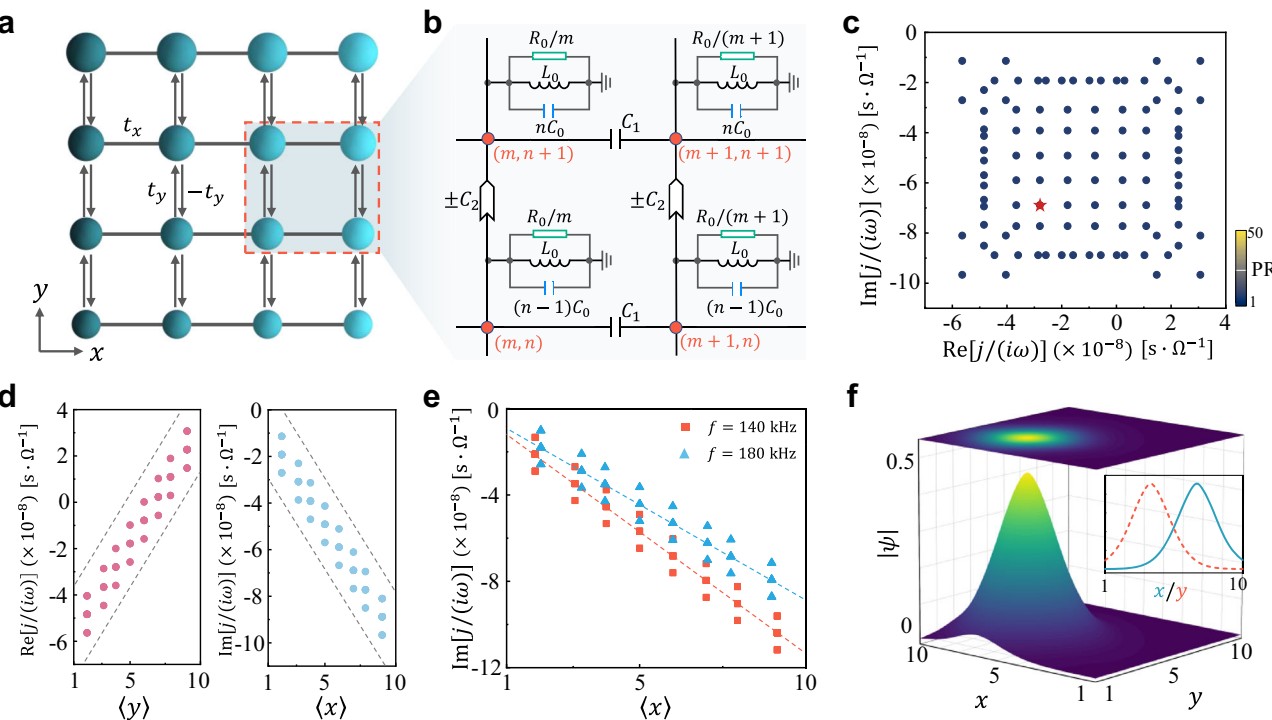

**Fig. 1 | Continuum Landau modes in a 2D non-Hermitian electric circuit.**
**a** Schematic of the 2D lattice with reciprocal hoppings $t_x$ along the $x$ direction (black lines), non-reciprocal hoppings $\pm t_y$ along the $y$ direction (black arrows), and complex linear on-site potential. The diameter and brilliant cyan of the circles denote the real ($nB_y$) and imaginary parts ($-imB_x$) of the complex linear on-site potential, respectively. **b** Schematic diagram of a part of the designed circuit network for the shaded rectangle in **a**. Each node (red dots) is connected to two adjacent nodes through capacitors $C_1$ along the $x$ direction and to two adjacent nodes through impedance converter with current inversion (INIC) of capacitance $\pm C_2$ along the $y$ direction. The nodes are further grounded by inductors $L_0$ as well as position-dependent capacitors $nC_0$ and resistors $R_0/m$, respectively. **c** Calculated admittance spectrum of the circuit Laplacian in Eq. (1) fed by the frequency $f = 162$ kHz. The

color of each point indicates the participation ratio of the corresponding eigenstate. The maximum value of the color bar denotes the PR of the admittance eigenstates without the linear on-site potential. **d** Plot of Re $[j/(i\omega)]$ (Left panel) and Im$[j/(i\omega)]$ (Right panel) versus the eigenstate's position expectation value $\langle y \rangle$ and $\langle x \rangle$, respectively. The gray-dashed lines denote the bounding lines obtained from $\mathcal{E}_{\mathbf{k+q}}^0$ in Eq. (2). **e** Calculated results of Im$[j/(i\omega)]$ versus $\langle x \rangle$ for the different frequencies $f = 140$ and $180$ kHz. The dashed lines denote the theoretical central trend lines ($\mathcal{E}_{\mathbf{k+q}}^0 \to 0$) obtained from Eq. (2). **f** The distribution of the eigenstate marked by star in **c**. The inset shows the amplitude distribution along lines passing through the center of the Gaussian envelope. In all subfigures, the other parameters are given by $C_0 = 10$ nF, $L_0 = 12.4$ μH, $R_0 = 100$ Ω, $C_1 = 10$ nF, and $C_2 = 10$ nF.

CLMs[37]. To clarify this point, we first consider a 2D Hermitian Dirac Hamiltonian with the same pseudovector potential **A**, i.e., $\mathcal{H}_{\mathbf{k}} = [0, J_{\mathbf{k}}/(i\omega); J_{\mathbf{k}}^*/(-i\omega), 0]$. In this case, the pseudomagnetic field gives rise to discrete Landau levels. In particular, the zeroth Landau level modes $\psi_0 = [0, Ce^{-\eta_x(x-x_0)^2} e^{-\eta_y(y-y_0)^2} e^{i\mathbf{q}\cdot\mathbf{r}}]^T$, where $\eta_x = -B_x/(2\mu_{\mathbf{k}})$, $\eta_y = B_y/(2\nu_{\mathbf{k}})$, $C$ is the normalized coefficient, and T denotes the transposition. These zeroth modes are the Gaussian wavepackets centered at $\mathbf{r}_0(\mathbf{k},\mathbf{q}) = (\mathrm{Im}(\mathcal{E}_{\mathbf{k}+\mathbf{q}}^0)/B_x, -\mathrm{Re}(\mathcal{E}_{\mathbf{k}+\mathbf{q}}^0)/B_y) + O(|\mathbf{q}|^2)$ for $B_x/\mu_{\mathbf{k}} < 0$ and $B_y/\nu_{\mathbf{k}} > 0$. For the non-Hermitian Dirac Hamiltonian described by $J_{\mathbf{k}}/(i\omega)$, its eigenstates $\psi_{\mathbf{k}}$ share the same Gaussian wavepackets as $\psi_0$, but with the displaced center position (see Supplementary Note 2 for details)

$$\mathbf{r}_0(j,\mathbf{k},\mathbf{q}) = \begin{pmatrix} x_0 \\ y_0 \end{pmatrix} = \begin{pmatrix} -\mathrm{Im}\left( j/(i\omega) - \mathcal{E}_{\mathbf{k}+\mathbf{q}}^0 \right)/B_x \\ \mathrm{Re}\left( j/(i\omega) - \mathcal{E}_{\mathbf{k}+\mathbf{q}}^0 \right)/B_y \end{pmatrix} + O(|\mathbf{q}|^2), \quad (2)$$

where $j/(i\omega)$ is the eigenvalue of $J_{\mathbf{k}}/(i\omega)$. Due to the slowly-varying envelope approximation, the solutions $\psi_0$ and $\psi_{\mathbf{k}}$ are limited in the regime $|\mathbf{q}| \ll 1$. Equation (2) shows that the eigenstate $\psi_{\mathbf{k}}$ with a given $j/(i\omega)$ can map to the zeroth Landau level mode of a given $\mathcal{H}_{\mathbf{k}}$ whose gauge is determined by $j/(i\omega)$. Since the gauge is continuous, the eigenvalues $j/(i\omega)$ can form continuous spectrum filling the complex energy plane. The eigenstates $\psi_{\mathbf{k}}$ are thus called CLMs.

To visualize the CLMs, we turn to the case of the finite lattice size. In this case, even though the eigenstates are finite and countable, the system remains the key property of the CLMs. Figure 1c shows the complex admittance spectrum of Eq. (1) associated with the participation ratio (PR) of the eigenstates (PR $= \sum_i |\psi_i|^2 / \sum_i |\psi_i|^4$) for $10 \times 10$ size. One can find that, as expected, the admittance eigenvalues form a finite area in the complex energy plane and all admittance eigenstates are localized as the small values of the PR[40]. By requiring $\mathbf{r}_0$ to lie in the circuit lattice, the boundaries of the admittance spectrum are given by $B_y + \epsilon_0 - 2|t_x| \le \mathrm{Re}[j/(i\omega)] \le B_y N_y + \epsilon_0 + 2|t_x|$ and $-B_x M_x - 2|t_y| \le \mathrm{Im}[j/(i\omega)] \le -B_x + 2|t_y|$, where $M_x$ and $N_y$ respectively denote the size of the circuit lattice in the $x$ and $y$ directions. These boundaries with the frequency-dependent parameters $\epsilon_0$ and $B_x$ shows clearly that the frequency causes the shift of both $\mathrm{Re}[j/(i\omega)]$ and $\mathrm{Im}[j/(i\omega)]$, while it only influences the bandwidth of $\mathrm{Im}[j/(i\omega)]$. The detailed numerical results are shown in Supplementary Fig. 3.

Figure 1d plots the real (imaginary) part of the admittance eigenvalues, $\mathrm{Re}[j/(i\omega)]$ ($\mathrm{Im}[j/(i\omega)]$), as a function of the eigenstate's position expectation values $\langle y \rangle$ ($\langle x \rangle$). It is evident that the center positions of the CLMs are linearly related to the complex eigenvalues $j/(i\omega)$, as suggested in Eq. (2). These linear relationships are given by $\mathrm{Re}[j/(i\omega)] = B_y y_0 + \epsilon_0$ and $\mathrm{Im}[j/(i\omega)] = -B_x x_0$ (see Supplementary Note 3 for details), which show that the frequency only introduces a shift constant in the linearity between $\mathrm{Re}[j/(i\omega)]$ and $y_0$ (Supplementary Fig. 4), while it can tune the slope of the linearity between $\mathrm{Im}[j/(i\omega)]$ and $x_0$. In Fig. 1e, we numerically plot $\mathrm{Im}[j/(i\omega)]$ versus the eigenstate's position expectation value $\langle x \rangle$ for the different frequencies $f = 140$ and $180$ kHz, respectively. These numerical results are consistent with the theoretical analysis.

Figure 1f shows the spatial distribution of a chosen admittance eigenstate marked by a star in Fig. 1c. A Gaussian-type wave function is clearly identified whose center position is consistent with the results in Fig. 1d. Similarly, the frequency can also affect the localization of the CLMs in the $x$ direction through the localization parameter $\eta_x$ of the Gaussian-type wave function (Supplementary Fig. 4). Since $\eta_x \propto 1/\omega$, this effect is very weak and is thus hard to be observed in experiments.

The roots of the admittance spectrum $j(\omega) = 0$ form the complex eigenfrequency spectrum of the system. This complex eigenfrequency spectrum has the same number of eigenstates as the complex admittance spectrum, and can thus form a continuum filling the complex frequency space, as shown in Supplementary Fig. 5. When the complex admittance or eigenfrequency spectra have a continuum, the voltage response is continuous, i.e., any frequency can excite the corresponding eigenmode of the circuit. For the complex eigenfrequency, its positive (negative) imaginary part indicates the dissipation (amplification) of the voltage. Since the oscilloscope could not capture the fast-changing dynamics of the exponentially oscillating amplitudes, it is hard to measure the complex eigenfrequency spectrum in experiments.

## Experiments in 2D circuit lattice

For the experimental realization of the CLMs, a circuit board containing $10 \times 10$ nodes was constructed. The circuit elements were pre-selected to show deviations less than 1% from their nominal values (see Methods for details). A photograph of a part of the circuit board is presented in Fig. 2a. The admittance eigenvalues and eigenstates are readily accessible by measuring the voltage response at each node to a local current input. Specifically, we measure the impedance matrix $G_{ab} = V_a/I_b$, where $V_a$ is the voltage response at any node $a$ in response to the local input current $I_b$ at node $b$. The complete matrix $G$ is the inverse of the circuit Laplacian $J(\omega)$ and therefore shares the same eigenvalues and eigenstates (see Supplementary Note 5 for details). In our experiments, all impedance measurements were performed with an impedance analyzer (Keysight E4990A).

In Fig. 2b, we measure the admittance spectrum together with PR in the complex energy plane at the frequency $f = 162$ kHz. It can be seen that the admittance eigenvalues are distributed over a finite region in the complex plane, and all eigenstates are localized. Note that for the clean circuit Laplacian considered in Eq. (1), the admittance spectrum exhibits a highly symmetrical pattern, as shown in Fig. 1c. In our experiment, the errors of the circuit components are about $\pm 1\%$. In this case, the admittance spectrum is cluttered, which agrees well with the simulated results in Supplementary Note 6.

In Fig. 2c, we measure the distribution of one localized admittance eigenstate marked in Fig. 2b. This admittance eigenstate exhibits a Gaussian-type envelope, which is consistent with the theoretical prediction (Fig. 1f). Figure 2d shows the experimental observations of $\mathrm{Re}[j/(i\omega)]$ and $\mathrm{Im}[j/(i\omega)]$ versus the eigenstate's position expectation values $\langle y \rangle$ and $\langle x \rangle$, respectively. This figure shows clearly that the center position of the CLMs is linearly related to the complex eigenvalue $j/(i\omega)$, as expected. As shown in Supplementary Fig. 8, when the errors of the circuit components increase to about $\pm 5\%$, the linear relationship still exists, which demonstrates the robustness of the CLMs. In Fig. 2e, we plot the measured results of $\mathrm{Im}[j/(i\omega)]$ versus $\langle x \rangle$ for the different frequencies $f = 140$ and $180$kHz. It shows that the frequency indeed affects the slope of the linearity between $\mathrm{Im}[j/(i\omega)]$ and $\langle x \rangle$. More experimental observations, including the frequency-dependent shift of the admittance spectrum and the linearity between $\mathrm{Re}[j/(i\omega)]$ and $\langle y \rangle$, are shown in Supplementary Fig. 9. These experimental observations are consistent with the calculated results in Figs. 1d, e.

Figure 2b–e demonstrates experimentally the existence of the CLMs. Note that the localized CLMs are centered at different positions, each localized CLM can thus be excited by feeding an a.c. current into the corresponding node of the circuit. Around the position of the excited node, the voltage response profile can exhibit a predominant weight. In experiments, the spatial feature of the CLMs can also be detected by injecting an a.c. current to excite one node of the circuit at its resonance frequency and then measuring the voltage response of all the circuit nodes. Here, we separately excite three nodes $(m,n) = (2,2)$, $(5,5)$, and $(9,9)$ for the different resonance frequencies $f_r = 197$, $179$, and $151$kHz. As shown in Fig. 2f, there is a dominant voltage signal at each excited node, demonstrating again the spatial feature of the CLMs. Note that the peak amplitude of the voltage response indicates the impedance of the node (i.e., $\mathrm{Im}[j/(i\omega)]$).

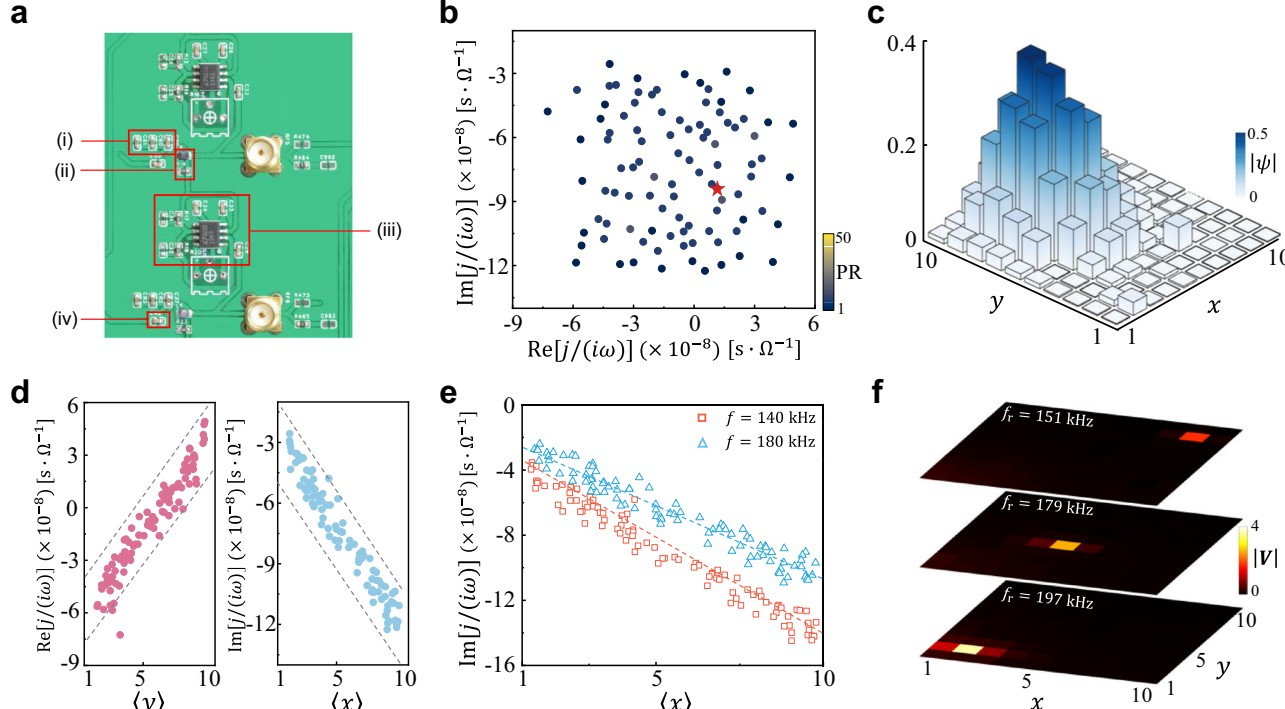

**Fig. 2 | Experimental observations of continuum Landau modes. a** Photograph of part of the circuit board, with numbers labeling the main circuit components: (i) the position-dependent capacitor $nC_0$ ($C_0 = 10$nF) built of parallel surface mounted device (SMD) capacitors; (ii) grounding for each node with an SMD inductor ($L_0 = 12.4$μH) and parallel SMD resistors yielding position-dependent resistor $R_0/m$ ($R_0 = 100$Ω); (iii) INIC made of an SMD capacitor ($C_2 = 10$nF), the operational amplifier LT1363 with supply voltages, and an equal pair of SMD resistor ($R_a = 1$kΩ); (iv) the reciprocal coupling built of SMD a capacitor ($C_1 = 10$nF). **b** Measured admittance spectrum for the frequency $f = 162$kHz. The color of each point indicates the participation ratio of the corresponding eigenstate. **c** Measured

distribution of the eigenstate marked by star in **b. d** Experimental observations of Re[$j/(i\omega)$] (Left Panel) and Im[$j/(i\omega)$] (Right Panel) versus the eigenstate's position expectation values $\langle y \rangle$ and $\langle x \rangle$, respectively. The gray-dashed lines are the corrected bounding lines by introducing loss offset and modified inductor. **e** Experimental observations of Im[$j/(i\omega)$] versus $\langle x \rangle$ for the different frequencies $f = 140$ and 180kHz. The dashed lines denote the corrected central trend lines ($\varepsilon_{k+q}^0 \to 0$) obtained from Eq. (2). **f** Measured voltage responses $|V|$ by separately exciting three positions (2,2), (5,5), and (9,9) at their resonance frequencies $f_r = 197$, 179, and 151kHz, respectively.

Interestingly, the linear relationship between the admittance spectrum and the localized position of its eigenstates manifests itself as a steady-state voltage response of full-field excitation. It should be noticed that the eigenmodes of the circuit correspond to the frequencies for which an exact eigenvalue Re$\left[ j/(i\omega) \right] = 0$ exists[41]. Accordingly, we calculate the resonance frequency spectrum of the circuit Laplacian by scanning the frequency. As shown in Fig. 3a, this resonance frequency spectrum is tilted due to the existence of the frequency-dependent coefficient $\epsilon_0$. Figure 3b shows the resonance frequency and Im$\left[ j/(i\omega) \right]$ of the resonance frequency spectrum versus the position expectation values $\langle y \rangle$ and $\langle x \rangle$, respectively. We can find that, in contrast to the admittance spectrum and eigenstates, the center position of the eigenmodes and the resonance frequency spectrum exhibit power-law behaviors, resulting from the coefficient $\epsilon_0 \sim 1/\omega^2$. In Fig. 3c, we show the steady-state voltage profile as a function of the frequency obtained from the LTSPICE simulation by feeding an a.c. current into all nodes of the circuit simultaneously. We find that the voltage accumulates at different nodes of the left edge of the circuit lattice for the different frequencies, i.e., position-frequency locking response at one side of the 2D circuit lattice. This is a consequence of the power-law relationship between Re$\left[ j/(i\omega) \right]$ and $\langle y \rangle$, causing a localized voltage response whose position is proportional to the excitation frequency in the $y$ direction. While the relation of Im$\left[ j/(i\omega) \right]$ and $\langle x \rangle$ implies the voltage response concentrating on the boundary with lowest relative loss in the $x$ direction. The combination of these two effects leads to such voltage response behavior of full-field excitation.

To observe the steady-state voltage of full-field excitation, multi-channel a.c. current feeds generated from a voltage source are injected

into all nodes through a shunt resistance ($R_s = 50$ Ω) separately. Then, we measure the voltage distribution on each node at different driving frequency. Figure 3d shows the detected voltage response versus frequency for the different nodes at the top ($n = 10$) and left ($m = 1$) circuit boundaries. For the nodes at the top boundary, the peak voltage amplitudes of all nodes are nearly frozen at a certain frequency. Moreover, the voltage is squeezed at the nodes with small grounding resistors, i.e., small values of $x$. As for the nodes at the left boundary, the frequency of the peak voltage response varies with the nodes $y$. Due to the frequency-dependent coefficient $\epsilon_0 \sim 1/\omega^2$, the frequency response spectrum broadens as $y$ decreases. These observations confirm the distinct response behaviors for the two directions of our circuit lattice, arising from the power-law relationship between admittance eigenstate center position and complex eigenvalues. The full voltage response as a function of frequency in the $x$-$y$ plane is displayed in Fig. 3e, showing good agreement with simulation results (Fig. 3c).

**Experiments in 1D circuit lattices**

In fact, the 1D circuit lattices described by the vertical or horizontal directions of Fig. 1a also exhibit the CLMs (see Supplementary Note 8 for details). Physically, it corresponds to the non-Hermitian Dirac Hamiltonian under the magnetic field with Landau gauge. To confirm this, we first consider the 1D lattice with non-reciprocal hoppings $\pm t_y$ and linear real on-site potential $nB_y$, whose circuit structure is shown in Fig. 4a. The circuit Laplacian is $J(\omega)/(i\omega) = \sum[t_y(|n\rangle\langle n+1| -$ H.c.$) + (nB_y + \epsilon_0)|n\rangle\langle n|]$ with $\epsilon_0 = - i/(\omega R_0) - 1/(\omega^2 L_0)$. In the continuum limit, it becomes $J_k/(i\omega) = i2t_y \sin k_y + \nu_k \partial_y + B_y y$, which amounts to imposing a pseudovector potential $\mathbf{A} = (-B_y y, 0)$ on the

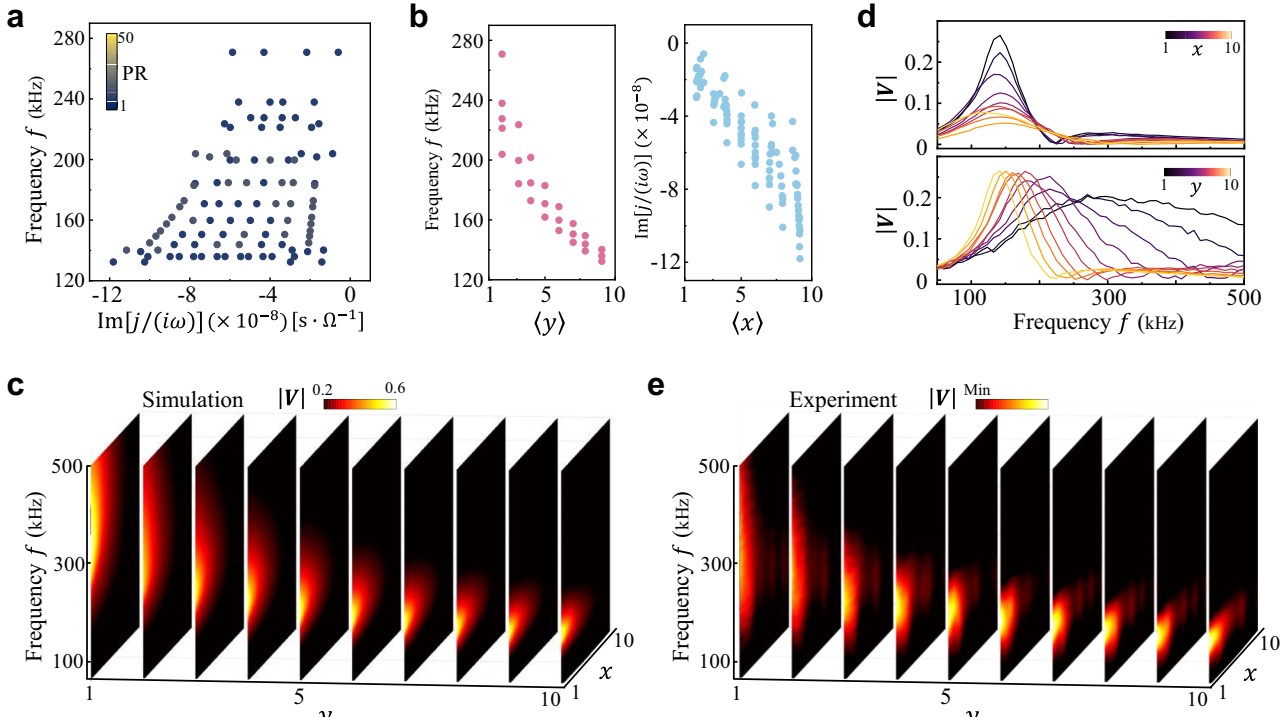

**Fig. 3 | Voltage response of the continuum Landau modes in 2D non-Hermitian electric circuit. a** Calculated resonance frequency spectrum of the circuit Laplacian. The color of each point indicates the participation ratio of the corresponding eigenstate. **b** Calculated resonance frequency (Left Panel) and Im[$j/(i\omega)$] (Right Panel) of the resonance frequency spectrum versus the eigenstate's position expectation values $\langle y \rangle$ and $\langle x \rangle$, respectively. **c** Simulated steady-state voltage response $|V|$ as a function of the frequency $f$ through injecting the a.c. current feed at all nodes simultaneously. **d** Measured steady-state voltage responses $|V|$ as functions of frequency for the different nodes in the top ($n = 10$, top panel) and left ($m = 1$, bottom panel) circuit boundaries. **e** Measured steady-state voltage response $|V|$ as a function of the frequency $f$ in the $x$-$y$ plane. In all subfigures, the parameters are the same as those in Fig. 2.

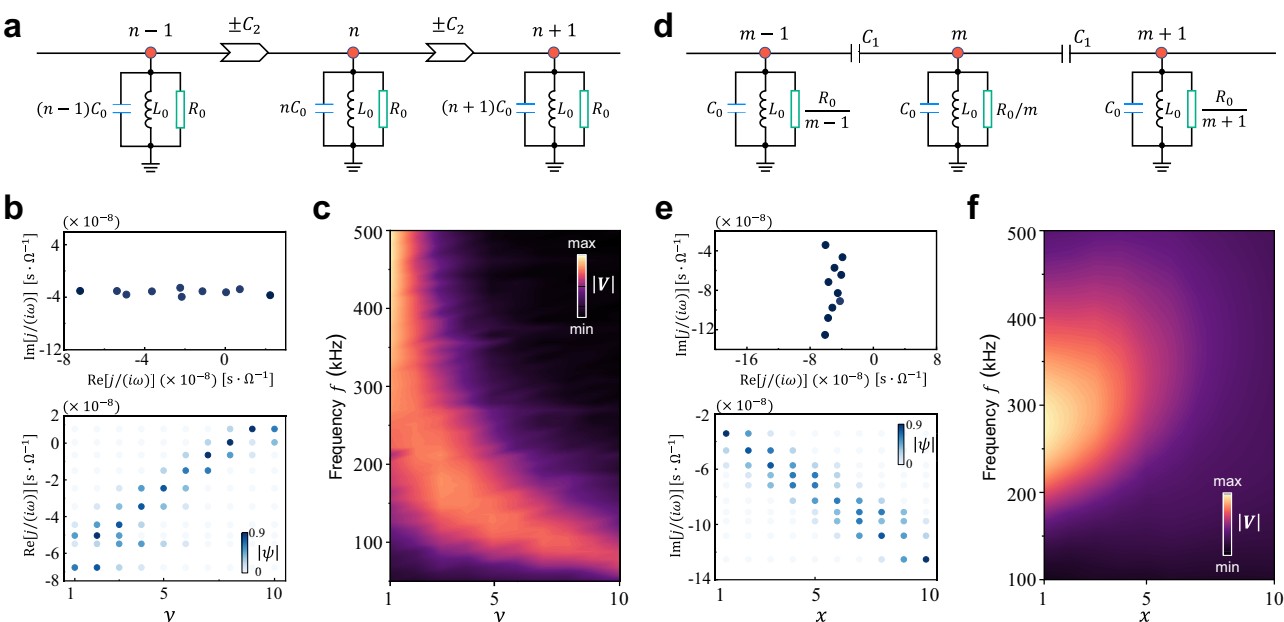

**Fig. 4 | Continuum Landau modes in 1D non-Hermitian electric circuits.**
**a** Schematic of the circuit chain with non-reciprocal hopping realized by INIC and linear real potential by position-dependent capacitors. **b** Measured admittance eigenvalues (top) and eigenstates (bottom) of the circuit chain in **a** for the frequency $f = 162$ kHz. **c** Steady-state voltage response versus the full-field excitation frequency for the circuit in **a**. **d** Schematic of the circuit chain with linear imaginary potential given by the position-dependent resistors. **e** Measured admittance eigenvalues (top) and eigenstates (bottom) of the circuit chain in **d** for the frequency $f = 162$ kHz. **f** Steady-state voltage response versus the full-field excitation frequency for the circuit in **d**. In subfigures **b**–**f**, the experimental parameters are the same as those in Fig. 2.

non-Hermitian Dirac Hamiltonian. The corresponding eigenstates are the CLMs centered at $y_0 = \mathrm{Re}[j/(i\omega)]/B_y$ (independent of $k_y$). This suggests that the system can be used as "rainbow traps" where the eigenstates are localized at positions proportional to eigenvalues[42–44]. We constructed a circuit chain containing 10 nodes to demonstrate this scenario. Figure 4b plots the measured complex admittance eigenvalues (top panel) along with the eigenstates (bottom panel) for the frequency $f = 162\,\mathrm{kH}$. The results agree with the theoretical predictions. The steady-state voltage response through full-field excitation is visualized in Fig. 4c. As expected, the voltage response has an amplitude peak positioned proportional to the excitation frequency.

Another 1D circuit lattice supporting the CLMs, depicted in Fig. 4d, has reciprocal hoppings $t_x$ and linear imaginary potential $-imB_x$. The corresponding circuit Laplacian is given by $J(\omega)/(i\omega) = \sum [t_x(|m\rangle\langle m+1| + \mathrm{H.c.}) + (\epsilon_0 - imB_x)|m\rangle\langle m|]$ with $\epsilon_0 = C_0 - 1/(\omega^2 L_0)$. In this case, we have $J_{\mathbf{k}}/(i\omega) = 2t_x \cos k_x + i\mu_{\mathbf{k}}\partial_x - iB_x x$, which is equivalent to introducing the Landau gauge $\mathbf{A} = (0, B_x x)$. The localized position of its eigenstates satisfies $x_0 = \mathrm{Im}[j/(i\omega)]/B_x$. Similarly, we plot the measured admittance eigenvalues and eigenstates for a circuit chain with 10 nodes and $f = 162\,\mathrm{kH}$, as presented in Fig. 4e. It is clear that the center positions of the eigenstates are linear related to the imaginary part of eigenvalues. Figure 4f shows the distribution of the steady-state voltage response. In contrast with the former case, the voltage response is concentrated at a boundary at which the modes with the lowest relative loss occupy. This implies that the system can act as a wave funnel with a finite bandwidth.

It should be noticed that for the current two cases of the 1D circuit lattices, the observed admittance eigenvalues in the complex plane are both localized around a distinct line (Fig. 4b and e). By tuning the parameters, the admittance eigenvalues can fill the complex plane (see Supplementary Fig. 12), in which the CLMs were predicted[37].

## Discussion

To summarize, we have employed non-Hermitian electric circuit networks to experimentally demonstrate the magnetic field-induced CLMs by measuring the admittance spectrum and eigenstates. Intriguingly, we have visualized an exotic steady-state voltage response through full-field excitation, resulting from the linear relationship between the CLMs' center position and complex eigenvalues. In particular, the voltage responses in 1D cases exhibit the behaviors of rainbow trapping or wave funneling. Our work provides a fertile platform to study rich non-Hermitian physics driven by the magnetic field[45–50].

Our experimental observations result from the non-Hermitian Dirac Hamiltonians with a uniform magnetic field. In this case, the Landau quantization and the edge states are missing. For the non-reciprocal model under a similar uniform magnetic field[48–50], the semiclassical trajectories of the wavepacket may turn out to be closed/skipping orbits in the 4D complex space[50]. The Landau levels exhibit the usual quantized spectra, and the Hall-like edge states are still found. In the circuits, the Landau levels can be measured through the admittance and impedance spectra, and the localized Landau modes and the Hall-like edge states can be observed by steady-state voltage response or dynamics of the excitation.

## Methods

### Circuit construction and measurements
The specific component values used for the non-Hermitian circuit implementation were pre-selected as $C_0 = 10\,\mathrm{nF}$ ($\pm 1\%$), $L_0 = 12.4\,\mu\mathrm{H}$ ($\pm 1\%$), $R_0 = 100\,\Omega$ ($\pm 1\%$), $C_1 = 10\,\mathrm{nF}$ ($\pm 1\%$), and $C_2 = 10\,\mathrm{nF}$ ($\pm 1\%$). For the implementation of the INIC, we used the unity-gain stable operational amplifier model LT1363.

To perform the measurements for the admittance eigenvalues and eigenstates, we measure the impedance matrix $G_{ab} = V_a/I_b$, where $V_a$ is the voltage response at any node $a$ in response to the local input current

$I_b$ at node $b$. The complete matrix $G$ is the inverse of the circuit Laplacian $J(\omega)$, and thus contains full information on admittance eigenvalues and eigenstates of the Laplacian. The impedance measurements were performed with an impedance analyzer (Keysight E4990A). For the measurements of steady-state voltage response through full-field excitation, multichannel a.c. current feeds generated from a voltage source are injected into all nodes through a shunt resistance separately. Then, we measure the voltage distributions on all nodes at different driving frequencies by using the oscilloscope (Keysight DSOX4024A).

## Data availability
The data that support the findings of this study are provided in the Source data file. Source data are provided with this paper.

## Code availability
Circuit simulations were performed using LTspice (https://www.analog.com/en/design-center/design-tools-and-calculators/ltspice-simulator.html#).

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

## Acknowledgements

This work is supported by the National Key R&D Program of China (Grant no. 2022YFA1404500), Cross-disciplinary Innovative Research Group Project of Henan Province (Grant no. 232300421004), and the National Natural Science Foundation of China (Grant nos. 12074232, 12125406, 12204290, 12374360, 12374312).

## Author contributions

G.C. and Z.W. conceived the idea. X.Z. and C.W. calculated the theoretical results, carried out the numerical simulations and performed the experiments. M.Y. guided the experimental measurement. N.L. calculated the theoretical results, and G.C. and Z.W. supervised the project. All the authors contributed to the preparation of the manuscript.

## Competing interests

The authors declare no competing interests.
