## [Peer Review File · Nature Communications]

REVIEWER COMMENTS

Reviewer #1 (Remarks to the Author):

This paper reports on an experimental realization of the recently-discovered phenomenon of Continuum Landau Modes (CLMs), using an electrical circuit lattice. This is a notable and significant work for the non-Hermitian physics community, since it demonstrates that non-Hermitian systems can violate the usual expectation that bound states (spatially localized eigenstates) must form a discrete spectrum.

The experiment sticks rather closely to the earlier theory paper [37], including in the design of the lattice and the set of experimental signatures. Much of the theoretical discussion from pages 3 to 5 recapitulate the discussions of the theory paper, though the text is succinct enough that this is acceptable. The new element is the implementation of the non-Hermitian Hamiltonian by means of a circuit Laplacian, which is based on previous developments in the field of circuit lattices [38,39].

The paper has a few shortcomings that ought to be corrected before it can be published. These are grouped into two categories.

First, it is advisable for the authors to do a slightly better job of articulating how their circuit implementation deviates from the theory paper [37]. There are a few places where there are interesting deviations, but they are not discussed prominently enough:

* In the circuit formulation, the frequency is a tuning parameter rather than an eigenvalue of the Hamiltonian. Since this was not anticipated in [37], the meaning of the frequency parameter (and particularly its effects on the CLMs) should be explained more clearly.

* Conversely, since the complex eigenvalues of the admittance matrix are not frequencies, the meaning of having a "continuum" of these quantities ought to be analyzed.

* The fact that the complex admittance eigenvalues can be directly measured in an experiment (Fig. 2b) is surely an important advantage of the circuit approach. Complex eigenfrequency spectra normally cannot be retrieved experimentally like this. Yet this was hardly emphasized in the text.

The second area where the paper falls short involves the key physical signatures of the 2D lattice, shown in Fig. 1f (numerical data) and Fig. 2f (experimental data).

* First of all, these subplots are too cramped, given their importance. The "3D slice" plots are hard to read, given that a lot of the data is obscured behind the slices. I would recommend splitting these subplots into a separate figure, or finding some other way of improving the presentation.

* Because of the role of the frequency in this system (as a tuning parameter), the significance of the frequency dependence of the voltage response is not obvious.

* A crucial feature of the CLMs is that they are spatially localized, with different CLMs centered at different positions. The voltage response in Fig. 1/2f does not seem to really capture this, as it mainly shows the voltage response being concentrated on the small-x side of the sample. Given the ability to probe the circuit in different ways, there should be a way for the CLMs' spatial characteristics to surface in a more convincing way, somehow. I would urge the authors to explore this.

If the above points can be satisfactorily addressed, I think this paper qualifies for publication in Nature Communications. The quality of the writing and figures is high, the technical aspects of the experiment seem to be excellent, and the topic is of fundamental interest for physics.

Reviewer #2 (Remarks to the Author):

The paper is theoretically sound and the experimental results are clearly presented. It is an interesting work which provides experimental demonstration of the consequences of a magnetic field analogue in a non-Hermitian Dirac fermion Hamiltonian, i.e., continuous Landau modes (CLM).

It constitutes the first demonstration of a non-Hermitian electrical circuit mimicking a physical magnetic field generating CLM. The effective magnetic field is constructed through the use of spatially varying imaginary on-site potentials to mimic the effects of the magnetic vector potential A .

The main achievement of the paper is in demonstrating a clear signature of a CLM analogue in an electrical circuit. This is no mean feat due to experimental issues like device tolerances and losses of the circuit elements. It also opens a new avenue of research into the effect of gauge terms in non-Hermitian systems, and its realization in actual physical/electrical circuits. As such, I am of the opinion that it is worthy of publication in Nature Comms.

However, could the authors comment on how other magnetic field induced/Landau level features in Hermitian circuits (e.g. skipping orbits) are transformed in non-Hermitian systems, and whether these can also give rise to electrical signatures in an electrical circuit.

Reviewer #3 (Remarks to the Author):

In the manuscript "Observation of continuum Landau modes in non-Hermitian electric circuits", the authors construct non-Hermitian electric circuit networks to simulate a non-Hermitian Dirac Hamiltonian under uniform magnetic field. They successfully capture the key characteristics of a recently identified novel state, referred to as the "Continuum Landau Modes (CLMs)", by measuring the admittance spectrum and eigenstates of these electric circuit networks. Furthermore, they observe phenomena akin to rainbow trapping or wave funneling in the voltage responses of 1D cases. To the best of my knowledge, this manuscript represents the first report of the experimental determination of the CLMs, an endeavor that is commendable. However, there are a few points that I believe the authors need to address before a final decision can be made.

i. There is a slight discrepancy between the parameters used in the calculations and simulations and those used in the experiment. This difference hinders an intuitive comparison between the results presented in Fig. 1 and Fig. 2. If it is not too time-consuming, I would recommend using the same parameters as those in the experimental setup for Fig. 1. These parameters also appear to be inconsistent with the content of the Methods section. The author should revise the manuscript to rectify errors that undermine its credibility.

ii. Moreover, the calculated spectra depicted in Fig. 1 (c) or Fig. 2 (b) and (c) of Ref. (37) exhibit highly symmetrical pattern. However, this characteristic is not observed in the measured admittance data. It would be beneficial if the author could make a simulation by LTspice or other similar software with/without consideration of components errors to investigate this discrepancy and provide a succinct discussion on the matter. We also notice the high-precision electronic components were utilized in experiment.

iii. In the section titled “Experiments in 1D Circuit Lattices”, the reference to Fig. 4 should be amended to Fig. 3.

iv. As shown in Fig. 3(b) and (e) of Ref. (37), ‘the eigenenergies fill the complex plane’ for the 1D case. However, the calculated results in the supplementary material, along with the experimental data, primarily show eigenenergies localized around a distinct line within the complex plane. What differentiates the model proposed by the authors from those described in Reference (37)?”

Reviewer #1 (Remarks to the Author):

This paper reports on an experimental realization of the recently-discovered phenomenon of Continuum Landau Modes (CLMs), using an electrical circuit lattice. This is a notable and significant work for the non-Hermitian physics community, since it demonstrates that non-Hermitian systems can violate the usual expectation that bound states (spatially localized eigenstates) must form a discrete spectrum.

The experiment sticks rather closely to the earlier theory paper [37], including in the design of the lattice and the set of experimental signatures. Much of the theoretical discussion from pages 3 to 5 recapitulate the discussions of the theory paper, though the text is succinct enough that this is acceptable. The new element is the implementation of the non-Hermitian Hamiltonian by means of a circuit Laplacian, which is based on previous developments in the field of circuit lattices [38,39].

Comment: The paper has a few shortcomings that ought to be corrected before it can be published. These are grouped into two categories.

Reply: We thank the referee for the positive comments of our work. In the revised manuscript, we have improved it significantly in terms of the referee's suggestions and comments. We hope that this improved version can be suitable for publication in Nature Communications.

Comment 1: First, it is advisable for the authors to do a slightly better job of articulating how their circuit implementation deviates from the theory paper [37]. There are a few places where there are interesting deviations, but they are not discussed prominently enough:

Reply 1: We thank the referee for the helpful comments. In the following reply and

revised manuscript, we have clarified the role of the frequency in our circuit system and demonstrated its effects on the CLMs. The continuum of the complex admittance and eigenfrequency spectra have been also analyzed.

Comment 2: In the circuit formulation, the frequency is a tuning parameter rather than an eigenvalue of the Hamiltonian. Since this was not anticipated in [37], the meaning of the frequency parameter (and particularly its effects on the CLMs) should be explained more clearly.

Reply 2: We thank the referee for the invaluable comment. Indeed, the frequency is a tuning parameter incorporated in the circuit Laplacian [Eq. (1) in the main text] with the coefficient $\epsilon_0(\omega)$ and the pseudomagnetic field $B_x(\omega)$. In the following, we illustrate the effects of these two frequency-dependent parameters on the admittance spectrum and eigenstates, respectively.

(i) The frequency modifies the admittance spectrum. We first note that, from the circuit Laplacian $J(\omega)/(i\omega)$, the diagonal components $\epsilon_0(\omega)$ and $B_x(\omega)$ induce the shift of the admittance spectrum along the directions of $\text{Re}[j/(i\omega)]$ and $\text{Im}[j/(i\omega)]$, respectively. Due to the existence of the site index m before $B_x(\omega)$, the bandwidth of $\text{Im}[j/(i\omega)]$ may vary with the frequency. On the other hand, by requiring the center position \mathbf{r}_0 of CLMs [Eq. (2) in the main text] to lie in the lattice, we can obtain the boundaries of the admittance spectrum as $B_y + \epsilon_0(\omega) - 2|t_x| \leq \text{Re}[j/(i\omega)] \leq B_y N_y + \epsilon_0(\omega) + 2|t_x|$ and $-B_x(\omega) M_x - 2|t_y| \leq \text{Im}[j/(i\omega)] \leq -B_x(\omega) + 2|t_y|$, where M_x and N_y denote the size of the circuit lattice in the x and y directions, respectively. It can be seen that $\epsilon_0(\omega)$ causes the shift of $\text{Re}[j/(i\omega)]$, while $B_x(\omega)$ not only gives rise to the shift of $\text{Im}[j/(i\omega)]$, but also influences its bandwidth.

Figures R1a and R1b show the complex admittance spectra for the frequencies $f = 162$ and 200 kHz, respectively. Obviously, the size and distributions of the admittance spectrum change with the frequency. We further plot the calculated results of $\text{Re}[j/(i\omega)]$ and $\text{Im}[j/(i\omega)]$ versus the frequency f , as shown respectively in

Figs. R1c and R1d. We can find that $\text{Re}[j/(i\omega)]$ varies with the frequency f and its bandwidth stays almost the same, as expected. While the bandwidth of $\text{Im}[j/(i\omega)]$ reduces with the increasing of the frequency. The experimental observations shown in Figs. R1e and R1f are in good agreement with theoretical ones.

(ii) The frequency affects the localization of the admittance eigenstates as well as the slope of the linearity between the CLM's center position and complex eigenvalues. On one hand, from the wavefunction ψ_0 of the CLM, the localization of the Gaussian envelope is characterized by $\eta_x = -B_x(\omega)/(2\mu_k)$ and $\eta_y = B_y/(2\nu_k)$. That is to say, the frequency can only affect the localization of the CLMs in the x direction. Figures R2a and R2b show the calculated and simulated amplitude distributions along lines passing through the center of one eigenstate (marked by star in Fig. R1) for the different frequencies, respectively. These calculated and simulated results agree well with the theoretical predictions. However, the effect ($\sim 1/\omega$) is very weak and is thus hard to be observed in experiment.

On the other hand, according to Eq. (2) in the main text, the linear relationships between complex admittance eigenvalues and the CLM's center position are given by $\text{Re}[j/(i\omega)] = B_y y_0 + \epsilon_0(\omega)$ and $\text{Im}[j/(i\omega)] = -B_x(\omega)x_0$. We can see that the frequency only introduces a shift constant in the linearity between $\text{Re}[j/(i\omega)]$ and y_0 , while it can tune the slope of the linearity between $\text{Im}[j/(i\omega)]$ and x_0 . In Figs. R2c and R2d, we show the real and imaginary parts of the admittance eigenvalues, $\text{Re}[j/(i\omega)]$ and $\text{Im}[j/(i\omega)]$, as functions of the eigenstate's position expectation values $\langle y \rangle$ and $\langle x \rangle$ for the different frequencies, respectively. The numerical results are consistent with the theoretical analysis. Figures R2e and R2f present the experimental results, which agree well with the theoretical ones.

In the revised manuscript, we have added new figures (Fig.1e and Fig.2e) and presented some related comments in Paragraphs 2 and 3 on Page 5. The detailed discussions have also been added in Supplementary S-III and S-VII.

Fig. R1 (a, b) Calculated admittance spectra of the circuit Laplacian [Eq. (1) in the main text] for the frequency $f = 162$ (a) and 200 (b) kHz. The color of each point indicates the participation ratio of the corresponding eigenstate. (c, d) Calculated results of $\text{Re}[j/(i\omega)]$ (c) and $\text{Im}[j/(i\omega)]$ (d) versus the frequency f . (e, f) Experimental observations of $\text{Re}[j/(i\omega)]$ (e) and $\text{Im}[j/(i\omega)]$ (f) versus the frequency f . In all subfigures, the parameters $C_0 = 10$ nF, $L_0 = 12.4$ μH , $R_0 = 100$ Ω , $C_1 = 10$ nF, $C_2 = 10$ nF, and $M_x = N_y = 10$.

Fig. R2 (a, b) Calculated (a) and simulated (b) amplitude distribution along lines passing through the center of one eigenstate (marked by star in Fig. R1a) for the different frequencies $f = 140, 160,$ and 180 kHz. The upper and lower panels indicate the results along the x and y directions, respectively. c Calculated results of $\text{Re}[j/(i\omega)]$ versus the eigenstate's position expectation value $\langle y \rangle$ for the different frequencies $f = 140$ and 180 kHz. d Calculated results of $\text{Im}[j/(i\omega)]$ versus the eigenstate's position expectation value $\langle x \rangle$ for the different frequencies $f = 140$ and 180 kHz. The dashed lines in c and d denote the theoretical central trend lines ($\mathcal{E}_{\mathbf{k}+\mathbf{q}}^0 \rightarrow 0$) obtained from Eq. (2) in the main text. (e, f) The experimental observations corresponding to (c, d) for the different frequencies $f = 140$ and 180 kHz, respectively. Here, the dashed lines are the corrected bounding lines by introducing the loss offset and modified inductor. In all subfigures, the other parameters are the same as those in Fig. R1.

Comment 3: Conversely, since the complex eigenvalues of the admittance matrix are not frequencies, the meaning of having a “continuum” of these quantities ought to be analyzed.

Reply 3: We thank the referee for the helpful suggestion. As an external parameter, the frequency in the circuit Laplacian can be tuned at will. For each frequency, the eigenvalues of the admittance matrix fill the complex energy plane if the lattice is infinite, i.e., they form a continuum. The roots of the admittance spectrum $j(\omega) = 0$ corresponds to the complex eigenfrequency spectrum of the system. As shown in Fig. R3, this complex eigenfrequency spectrum has the same number of the eigenstates as the complex admittance spectrum, and can thus form a continuum filling the complex frequency space. When the complex admittance or eigenfrequency spectra form a continuum, the voltage response is continuous, i.e., any frequency can excite the corresponding eigenmode of the circuit.

These discussions have been added in Paragraph 2 on Page 6 of the revised manuscript.

Fig. R3 **a** Calculated admittance spectrum of the circuit Laplacian in Eq. (1) fed by the frequency $f = 162$ kHz. The color of each point indicates the participation ratio of the corresponding eigenstate. **b** Complex eigenfrequency spectrum (f') close to the frequency $f = 162$ kHz. The other parameters are the same as those in Fig. R1.

Comment 4: The fact that the complex admittance eigenvalues can be directly

measured in an experiment (Fig. 2b) is surely an important advantage of the circuit approach. Complex eigenfrequency spectra normally cannot be retrieved experimentally like this. Yet this was hardly emphasized in the text.

Reply 4: We thank the referee for the comment. For the complex eigenfrequency, its positive (negative) imaginary part indicates the dissipation (amplification) of the voltage. Since the oscilloscope could not capture the fast-changing dynamics of the exponentially oscillating amplitudes, it is hard to measure the complex eigenfrequency spectrum in experiments. However, the admittance matrix can be experimentally reconstructed by measuring the voltage response at each node to a local a.c. input. Moreover, the admittance eigenvalues and eigenstates can be extracted through numerical diagonalization.

In the revised manuscript, we have emphasized the above point in Paragraph 2 on Page 6.

Comment 5: The second area where the paper falls short involves the key physical signatures of the 2D lattice, shown in Fig. 1f (numerical data) and Fig. 2f (experimental data).

Reply 5: We thank the referee for the valuable suggestion. In the revised manuscript, we have extended key physical signatures of the 2D lattice, including replotting Figs. 1f and 2f of the original manuscript (Figs. 3c and 3e of the revised manuscript), illustrating the significance of the frequency dependence of the voltage response, and redesigning experimental method to observe the CLMs' spatial characteristics.

Comment 6: First of all, these subplots are too cramped, given their importance. The “3D slice” plots are hard to read, given that a lot of the data is obscured behind the slices. I would recommend splitting these subplots into a separate figure, or finding some other way of improving the presentation.

Reply 6: We thank the referee for the constructive suggestion. In the revised manuscript, we have enlarged the spacing between the 3D slices of the Fig. 1f and Fig. 2f in the original manuscript to better show all the data. These two subfigures have been reorganized into Fig. 3 of the revised manuscript.

Comment 7: Because of the role of the frequency in this system (as a tuning parameter), the significance of the frequency dependence of the voltage response is not obvious.

Reply 7: We thank the referee for the comment. To show the frequency dependence of the voltage response, we investigate the resonance frequency spectrum of the circuit Laplacian by scanning the frequency. The resonance frequency spectrum is obtained by $\text{Re}[j/(i\omega)] = 0$, corresponds to eigenmodes of the circuit. As shown in Fig. R4a, the resonance frequency spectrum is tilted due to the existence of the frequency-dependent coefficient $\epsilon_0(\omega)$. The eigenmodes share the same size (number of modes) as the admittance eigenstates and are also localized. Figure R4b shows the resonance frequency and $\text{Im}[j/(i\omega)]$ of the resonance frequency spectrum versus the position expectation values $\langle y \rangle$ and $\langle x \rangle$, respectively. We can find that, in contrast to the admittance spectrum and eigenstates, the center position of the eigenmodes and resonance frequency spectrum exhibit power-law behaviors, which result from the coefficient $\epsilon_0(\omega) \sim 1/\omega^2$. The position-frequency locking, together with the boundary-localized eigenmodes with the lowest relative loss in the x direction, lead to the unique voltage response, as shown in Figs. 3e and 3f of the revised manuscript. The simulated and experimental results of the voltage response are consistent with the theoretical analysis.

In the revised manuscript, we have added Figs. 3a and 3b, together with the related discussions in Paragraph 1 on Page 8, to explain the frequency dependence of the voltage response.

Fig. R4 **a** Calculated resonance frequency spectrum of the circuit Laplacian. The color of each point indicates the participation ratio of the corresponding eigenstate. **b** Calculated resonance frequency (Left Panel) and $\text{Im}[j/(i\omega)]$ (Right Panel) of the resonance frequency spectrum versus the eigenstate's position expectation values $\langle y \rangle$ and $\langle x \rangle$, respectively. In all subfigures, the parameters are the same as those in Fig. R1.

Comment 8: A crucial feature of the CLMs is that they are spatially localized, with different CLMs centered at different positions. The voltage response in Fig. 1/2f does not seem to really capture this, as it mainly shows the voltage response being concentrated on the small- x side of the sample. Given the ability to probe the circuit in different ways, there should be a way for the CLMs' spatial characteristics to surface in a more convincing way, somehow. I would urge the authors to explore this.

Reply 8: We thank the referee for the suggestion. Since the localized CLMs are centered at different positions, each localized CLM can be excited by feeding an a.c. current into the corresponding node of the circuit. Around the position of the excited node, the voltage response profile can exhibit a predominant weight. In experiments, we inject an a.c. current to excite one node of the circuit at its resonant frequency and then measure the voltage response of all the circuit nodes to show the spatial characteristics of the CLMs. Here we separately excite three nodes $(m, n) = (2, 2)$, $(5, 5)$, and $(9, 9)$ for the different resonance frequencies. As shown in Fig. R5, there is a dominant voltage signal at each excited node, demonstrating clearly the spatial feature of the CLMs (different CLMs are centered at different positions). Note that the peak amplitude of the voltage response indicates the impedance of the node (i.e.,

$\text{Im}[j/(i\omega)]$.

In the revised manuscript, we have added Fig. 2f and related discussions in Paragraph 3 on Page 7.

Fig. R5 Measured voltage response $|V|$ by exciting three positions (2,2)(a), (5,5) (b), and (9,9) (c) at their resonance frequencies $f_r = 197$, 179, and 151 kHz, respectively.

If the above points can be satisfactorily addressed, I think this paper qualifies for publication in Nature Communications. The quality of the writing and figures is high, the technical aspects of the experiment seem to be excellent, and the topic is of fundamental interest for physics.

Reply: We thank the referee for the positive evaluation of our work. In the revised manuscript, we have addressed all suggestions and comments raised by the referee and improved it significantly. We hope that the referee can recommend our revised manuscript to be published in Nature Communications.

Reviewer #2 (Remarks to the Author):

The paper is theoretically sound and the experimental results are clearly presented. It is an interesting work which provides experimental demonstration of the consequences of a magnetic field analogue in a non-Hermitian Dirac fermion Hamiltonian, i.e., continuous Landau modes (CLM).

It constitutes the first demonstration of a non-Hermitian electrical circuit mimicking a physical magnetic field generating CLM. The effective magnetic field is constructed through the use of spatially varying imaginary on-site potentials to mimic the effects of the magnetic vector potential A .

The main achievement of the paper is in demonstrating a clear signature of a CLM analogue in an electrical circuit. This is no mean feat due to experimental issues like device tolerances and losses of the circuit elements. It also opens a new avenue of research into the effect of gauge terms in non-Hermitian systems, and its realization in actual physical/electrical circuits. As such, I am of the opinion that it is worthy of publication in Nature Comms.

Reply: We thank the referee for the very positive comments of our work.

Comment: However, could the authors comment on how other magnetic field induced/Landau level features in Hermitian circuits (e.g. skipping orbits) are transformed in non-Hermitian systems, and whether these can also give rise to electrical signatures in an electrical circuit.

Reply: We thank the referee for the inspiring question. What happens to the other magnetic field-induced features in the non-Hermitian case depends crucially on the specific models of the systems. Here we cannot give a clear and definite answer. For

the present studies about the interplay of non-Hermiticity and magnetic field, we present three comments as follows.

(i) For the non-Hermitian Dirac Hamiltonians under a magnetic field considered by our work, the eigenstates of the system have Gaussian spatial envelopes and form a continuum filling the complex energy plane. These modes can map to the zeroth Landau level modes of the Hermitian Dirac Hamiltonian. Therefore, the Landau quantization and the edge states (i.e., skipping orbits in semi-classical view) are missing.

(ii) For the non-reciprocal model under a magnetic field (see, for example, References [46-47, 49-50] of the revised manuscript), the semiclassical trajectories of the wavepacket may turn out to be closed/skipping orbits in the 4D complex space [47]. The Landau levels exhibit the usual quantized spectra and the Hall-like edge states are still found.

(iii) Benefiting from a wide range of passive and active circuit elements in electric circuit networks, it is interesting to experimentally investigate these unique non-Hermitian effects controlled by magnetic field. For example, the pseudomagnetic field can be mimicked through inhomogeneous strain of the graphene-like circuit lattice and the non-Hermitian terms can be achieved by choosing appropriate circuit elements. The Landau levels can be measured through admittance and impedance spectra. The localized Landau modes and the Hall-like edge states can be observed by steady-state voltage response or dynamics of the excitation.

In Paragraph 4 on Page 10 of the revised manuscript, we have added some discussions and outlook about other magnetic field induced effects controlled by non-Hermiticity.

Reviewer #3 (Remarks to the Author):

In the manuscript “Observation of continuum Landau modes in non-Hermitian electric circuits”, the authors construct non-Hermitian electric circuit networks to simulate a non-Hermitian Dirac Hamiltonian under uniform magnetic field. They successfully capture the key characteristics of a recently identified novel state, referred to as the “Continuum Landau Modes (CLMs)”, by measuring the admittance spectrum and eigenstates of these electric circuit networks. Furthermore, they observe phenomena akin to rainbow trapping or wave funneling in the voltage responses of 1D cases. To the best of my knowledge, this manuscript represents the first report of the experimental determination of the CLMs, an endeavor that is commendable. However, there are a few points that I believe the authors need to address before a final decision can be made.

Reply: We thank the referee for the positive comments of our work. In the revised manuscript, we have improved it significantly in terms of the referee’s suggestions and comments. We hope the referee can recommend our revised manuscript to be published in Nature Communications.

Comment 1: There is a slight discrepancy between the parameters used in the calculations and simulations and those used in the experiment. This difference hinders an intuitive comparison between the results presented in Fig. 1 and Fig. 2. If it is not too time-consuming, I would recommend using the same parameters as those in the experimental setup for Fig. 1. These parameters also appear to be inconsistent with the content of the Methods section. The author should revise the manuscript to rectify errors that undermine its credibility.

Reply 1: We thank the referee for the helpful suggestion. In the revised manuscript, we have replotted Fig. 1 according to the experimental parameters (i.e., those in Fig. 2). In addition, we have also corrected the parameters in the Method section.

Comment 2: Moreover, the calculated spectra depicted in Fig. 1(c) or Fig. 2(b) and (c) of Ref. (37) exhibit highly symmetrical pattern. However, this characteristic is not observed in the measured admittance data. It would be beneficial if the author could make a simulation by LTspice or other similar software with/without consideration of components errors to investigate this discrepancy and provide a succinct discussion on the matter. We also notice the high-precision electronic components were utilized in experiment.

Reply 2: We thank the referee for the nice suggestion. In Fig. 1c or Figs. 2b and 2c of Reference (37), the clean circuit Laplacian/Hamiltonian is considered, and the admittance/energy spectra are thus highly symmetric. In the real experiments, the errors of the electronic components usually exist. For example, in our experiment with the high-precision electronic components, these errors are about $\pm 1\%$. In order to investigate the effects induced by the errors, we use the LTspice software to simulate the admittance spectra and the eigenstate's energy against the position expected values by introducing $\pm 1\%$ (a, b) and $\pm 5\%$ (c, d) disorders to all the circuit components. As shown in Fig. R6a, the admittance spectrum turns out to be cluttered, even if the high-precision electronic components with the errors of $\pm 1\%$ are introduced. However, the key features of the CLMs, the localization of the eigenstates and the linear relationship between the eigenstates' center position and eigenvalues, still exist (Fig. R6b). When the errors increase ($\pm 5\%$), the similar properties of the admittance spectra and the linear relationship between the eigenstates' center position and eigenvalues are found (Fig. R6c and R6d), which means that the features of the CLMs are robust to the errors of the electronic components.

We have pointed out the discrepancy of the experimental results of the admittance spectrum in Paragraph 2 on Page 7 of the revised manuscript. The related discussions and simulation results have been added in Supplementary S-VI.

Fig. R6 **a** Simulated admittance spectrum of the circuit Laplacian by considering the error of $\pm 1\%$ for all circuit components. The color of each point indicates the participation ratio of the corresponding eigenstate. **b** Simulated results of $\text{Re}[j/(i\omega)]$ (left panel) and $\text{Im}[j/(i\omega)]$ (right panel) versus the expectation values of the eigenstate's position, $\langle y \rangle$ and $\langle x \rangle$, respectively. **c**, **d** The simulated results corresponding to **a** and **b** with the error of $\pm 5\%$ for all circuit components. The other parameters are the same as those in Fig. 2 of the revised manuscript.

Comment 3: In the section titled “Experiments in 1D Circuit Lattices”, the reference to Fig. 4 should be amended to Fig. 3.

Reply 3: We thank the referee for pointing out the typo. In the revised manuscript, we have checked the reference according to the replotted figures.

Comment 4: As shown in Fig. 3(b) and (e) of Ref. (37), ‘the eigenenergies fill the complex plane’ for the 1D case. However, the calculated results in the supplementary material, along with the experimental data, primarily show eigenenergies localized

around a distinct line within the complex plane. What differentiates the model proposed by the authors from those described in Reference (37)?”

Reply 4: We thank the referee for the nice question. The circuit Laplacian of our circuit is an analog of the model in Reference (37). The additional parameter ϵ_0 in the circuit model just induces a frequency-dependent shift of the complex admittance spectrum.

The distributions of the admittance spectra (line or finite area) of our 1D circuit lattices depend on the magnitude of the pseudomagnetic field (i.e., C_0 or R_0). In Figs. R7a-c, we plot the admittance spectrum of the 1D circuit lattice in Fig. 4a of the revised manuscript for $C_0 = 1$ pF (a), $C_0 = 50$ pF (b), and $C_0 = 10$ nF (c), respectively. One can see that when $C_0 = 50$ pF, the admittance spectrum can fill the complex plane (Fig. R7b), similar to Fig. 3(b) of Ref. (37), and it becomes a line for $C_0 = 1$ pF (Fig. R7a) or $C_0 = 10$ nF (Fig. R7c). The similar distributions of the admittance spectra can also be found in the circuit lattice in Fig. 4d of the revised manuscript for different $R_0 = 350$ Ω (Fig. R7d), $R_0 = 10$ k Ω (Fig. R7e), and $R_0 = 100$ k Ω (Fig. R7f). In our experiments, we choose the parameters $C_0 = 10$ nF and $R_0 = 100$ Ω , in which a linear distribution of the admittance spectrum in the complex plane is considered. It should be emphasized that this linear distribution does not affect the key feature of the CLMs (i.e., the CLMs' center position varies linearly with the complex admittance eigenvalues), as shown in Fig. 4 of the revised manuscript.

The above discussions have been added in Paragraph 2 on Page 10 and Supplementary S-VIII of the revised manuscript.

Fig. R7 **a-c** Admittance spectra of the circuit Laplacian in Fig. 4a for $C_0 = 1$ pF (**a**), $C_0 = 50$ pF (**b**), and $C_0 = 10$ nF (**c**). The color of each point denotes the eigenstate's position expectation value $\langle y \rangle$. **d-f** Admittance spectra of the circuit Laplacian in Fig. 4d for $R_0 = 350 \Omega$ (**d**), $R_0 = 10$ k Ω (**e**), and $R_0 = 100$ k Ω (**f**). The color of each point denotes the eigenstate's position expectation value $\langle x \rangle$. In all subfigures, the size of the circuit lattice is chosen as 1000. The other parameters are the same as those in Fig. 4 of the revised manuscript.

REVIEWERS' COMMENTS

Reviewer #1 (Remarks to the Author):

The authors have done a good job of addressing the points raised during the first reviewing round. I have no further substantive criticisms of the paper, and can recommend it for publication in Nature Communications. Prior to publication, I recommend that the authors do another round of checking for grammar and spelling mistakes; there are some errors which should be easy to spot using a computer checker.

Reviewer #2 (Remarks to the Author):

In my opinion, the authors have satisfactorily addressed the comments by me and the other reviewers. I recommend publication of the paper in Nature Communications.

Reviewer #3 (Remarks to the Author):

After reading all the reviewers' comments, the authors' reply and the revised manuscript, I think the authors have well answered all the points and revised the manuscript accordingly. Thus, I recommend it to be published in Nature Communications.

Reviewer #1 (Remarks to the Author):

Comment: The authors have done a good job of addressing the points raised during the first reviewing round. I have no further substantive criticisms of the paper, and can recommend it for publication in Nature Communications. Prior to publication, I recommend that the authors do another round of checking for grammar and spelling mistakes; there are some errors which should be easy to spot using a computer checker.

Reply: We thank Reviewer for the recommendation. In the revised manuscript, we have carefully checked and corrected the grammar and spelling mistakes.

Reviewer #2 (Remarks to the Author):

Comment: In my opinion, the authors have satisfactorily addressed the comments by me and the other reviewers. I recommend publication of the paper in Nature Communications.

Reply: We thank the reviewer for recommending the publication of our manuscript.

Reviewer #3 (Remarks to the Author):

After reading all the reviewers' comments, the authors' reply and the revised manuscript, I think the authors have well answered all the points and revised the manuscript accordingly. Thus, I recommend it to be published in Nature Communications.

Reply: We thank the reviewer for recommending the publication of our manuscript.